# Cryo- EM structure of the mycobacterial 70S ribosome in complex with ribosome hibernation promotion factor RafH

Niraj Kumar [1], Shivani Sharma[1] & Prem S. Kaushal [1] ✉

Ribosome hibernation is a key survival strategy bacteria adopt under environmental stress, where a protein, hibernation promotion factor (HPF), transitorily inactivates the ribosome. *Mycobacterium tuberculosis* encounters hypoxia (low oxygen) as a major stress in the host macrophages, and upregulates the expression of RafH protein, which is crucial for its survival. The RafH, a dual domain HPF, an orthologue of bacterial long HPF (HPF[long]), hibernates ribosome in 70S monosome form, whereas in other bacteria, the HPF[long] induces 70S ribosome dimerization and hibernates its ribosome in 100S disome form. Here, we report the cryo- EM structure of *M. smegmatis*, a close homolog of *M. tuberculosis*, 70S ribosome in complex with the RafH factor at an overall 2.8 Å resolution. The N- terminus domain (NTD) of RafH binds to the decoding center, similarly to HPF[long] NTD. In contrast, the C-terminus domain (CTD) of RafH, which is larger than the HPF[long] CTD, binds to a distinct site at the platform binding center of the ribosomal small subunit. The two domain-connecting linker regions, which remain mostly disordered in earlier reported HPF[long] structures, interact mainly with the anti-Shine Dalgarno sequence of the 16S rRNA.

Protein synthesis or translation is a vital cellular process that occurs on ribosomes in all cells and consumes nearly half of the cell's resources[1–4]. When bacteria encounter unfavorable conditions and cease to grow, the rate of protein synthesis is regulated by reducing de novo ribosome synthesis[5], degradation of excess ribosome[6,7], and rapid modulation or inhibition of existing ribosomes by a variety of factors[8]. The widely adopted mechanism of ribosome modulation is ribosome hibernation, wherein a protein factor, hibernation promotion factor (HPF), reversibly binds to the ribosome and stabilizes it in an inactive hibernating state[9,10]. Ribosome hibernation is a highly conserved, tightly regulated process in bacteria and is responsible for the survival of growth-arrested bacterial cells under environmental stresses in a drug-tolerant state[9,11]. Under nutrition starvation, the free ribosomal subunits become more prone to ribonuclease degradation[12], particularly the 3′ end of the 16S rRNA, which harbors the anti-Shine Dalgarno sequence[13] has been reported as the target site

for the 3′ to 5′ exoribonucleases[12,14,15]. In a *Δhpf* strain of *E. coli*, the 16S rRNA is degraded by fragmenting at specific sites and trimming its 3′ end[15]. *Δhpf Staphylococcus aureus* strain showed reduced virulence in a murine model of infection[16], and its ribosome becomes extremely sensitive to nucleolytic cleavage[17].

The process of ribosome hibernation is well studied in enteric bacteria (Supplementary Fig. 1), which possess mainly two forms of HPFs[9,10]. The HPF long (HPF[long]), a two domain protein factor, is found in most gram-positive bacteria and is solely responsible for inducing the 100S ribosome formation (Supplementary Fig. 1a). The HPF[long] induces 100S disome formation through dimerization of its C-terminus domain. The HPF short (HPF[short]), a single domain protein, induces 100S ribosome (disome) formation with another factor known as ribosome modulation factor (RMF) mainly found in *E. coli* and other γ-proteobacteria (Supplementary Fig. 1b). The molecular mechanism of 100S ribosome hibernation is thoroughly studied[9], and the cryo- EM

[1]Structural Biology & Translation Regulation Laboratory, UNESCO-DBT, Regional Centre for Biotechnology, NCR Biotech Science Cluster, Faridabad 121 001, India. ✉e-mail: prem.kaushal@rcb.res.in

structures of hibernating 100S ribosomes from different bacterial species are available[18–24]. Another mode of ribosome hibernation is induced by a single domain protein, the YfiA, also known as protein Y (encoded by gene *yfiA*, also known as *raiA*)[25] and its orthologue in chloroplast ribosome is known as PSRP-1[26,27] which hibernates ribosome in the 70S (monosome) form only (Supplementary Fig. 1c).

*Mycobacterium tuberculosis* (Mtb), the causative agent of one of the deadliest diseases, tuberculosis (TB), is also capable of maintaining a dormant stage in the hostile environment of host macrophages causing Latent Tuberculosis Infection (LTBI)[28–31]. The LTBI is known to exist in one quarter of the world's population[32–34], where the pathogen down regulates the vast majority of the metabolic processes, thus imparting resistance to various antibiotics[35] and serving as a vast reservoir for TB infection[36]. One such significantly affected process is translation, which is also a target for nearly 40% of known antibiotics[37]. Overall, the translation machinery in mycobacteria is conserved and possesses unique structural features associated with its ribosome architecture[11,38–45] such as H54a, a ~110 nucleotide insertion in H54 of the 23S rRNA[39–41]. Another distinctive feature associated with it is, its ribosome hibernation, which has been proposed to be a primary survival mechanism for non-replicating Mtb[11,42]. Mycobacteria hibernates ribosomes in 70S monosome form only, any higher order ribosome structure, such as 100S disome, has not been reported so far[38,42,43] (Supplementary Fig. 1d, e).

Mycobacterial HPF, the mycobacterial protein Y (MPY), (also designated as a ribosome associated factor under stasis RafS)[38] induces 70S ribosome hibernation (Supplementary Fig. 1d) under different environmental stress, such as carbon starvation[38], zinc starvation[42], and in stationary phase[43]. The MPY possesses two domains and a connecting linker region, HPF[long] like organization (Supplementary Fig. 1d). Its CTD and linker region remain disordered in reported structures[42,43]. Thus, its binding site information and the structural basis of MPY's inability to induce ribosome dimerization to form 100S remains unknown.

Mycobacterium contains another HPF known as RafH, and its expression is upregulated under hypoxia (low oxygen) stress through DosR regulon[46]. Mtb encounters multiple stresses, primarily the hypoxia, in host macrophages[47]. Hypoxia induces the DosR regulon, which upregulates nearly 48 genes, including RafH (ribosome associated factor under hypoxia)[38] expressing gene MSMEG_3935 in *M. smegmatis* and Rv0079 in *M. tuberculosis*[48]. RafH appears to be the major factor responsible for Mtb's survival under hypoxia stress and promotes cellular viability in a growth-arrested state[38]. A *ΔdosR M. smegmatis* strain showed significant levels of rRNA degradation compared with the wild-type strain, and the *ΔdosR* phenotype gets alleviated by adding an extra copy of *rafH* gene[38]. Overexpression of the RafH factor led to an early entry to the stationary phase in *E. coli*, and its gene was found to be conserved in many clinical isolates[48]. RafH is a dual domain HPF, an orthologue of the HPF[long], but still cannot induce ribosome dimerization, and stabilizes ribosome in the associated 70S form ref. [38] (Supplementary Fig. 1e). The structural basis of RafH induced ribosome hibernation, and its inability to form 100S like disome is unknown, as no structure is available.

Here, we report the single particle cryo-EM structure of *M. smegmatis* (a close homolog of *M. tuberculosis*) 70S ribosome in complex with RafH at an overall 2.8 Å resolution. In addition, we also report 70S ribosome in complex with RafH and bS1 ribosomal (r−) protein and 70S ribosome in complex with RafH and E-site tRNA, both cryo-EM maps low pass filtered at 3.5 Å resolution. The structure reveals that RafH NTD binds to a conserved binding site at the small subunit decoding center. In contrast, RafH CTD binds to a unique position at the small subunit platform binding center, which has not been reported before. The linker region connecting two domains interacts primarily with the anti-Shine Dalgarno (a-SD) sequence of the 16S rRNA. Intriguingly, the study reports this remarkable interaction

between the HPF linker and a-SD in atomic details, and reveals the structural basis for mycobacteria's inability to form 100S like hibernating ribosomes.

## Results

### 70S ribosome RafH complex formation and protein synthesis inhibition

The 70S ribosomes were purified by sucrose density gradient ultracentrifugation (Fig. 1a). To remove co-purified translation protein factors, mRNA and tRNAs, the 70S ribosomes were dissociated into their respective subunits by lowering the $MgCl_2$ to 1 mM (Fig. 1b) and further re-associated by incubating equimolar concentrations of 50S and 30S subunits in 20 mM $MgCl_2$ (Fig. 1c). The 70S ribosome RafH complex, prepared by mixing re-associated 70S ribosome with purified RafH protein, was confirmed by sucrose pelleting assay (Fig. 1d). The RafH protein band was visible in SDS-PAGE for the pellet fraction of the 70S ribosome RafH reaction mixture, suggesting RafH binds to the 70S ribosome. As expected, the corresponding band was absent in the pellet fraction of ribosome without RafH (Fig. 1d). Similarly, the RafH protein band was clearly visible in pellet fraction of the 30S ribosome on SDS-PAGE, indicating that RafH binds to the 30S ribosome subunit as well (Supplementary Fig. 2).

In-vitro translation assay, performed by titrating ribosomes to RafH with different stoichiometry ratios of 1:0, 1:1, and 1:2, showed that RafH inhibits the protein synthesis (Fig. 1e), which also confirmed that the purified protein was in an active conformation. The RafH point mutant W96A shows slightly lesser inhibition of protein synthesis, whereas another RafH point mutant W111A shows similar inhibition as compared to the wild-type RafH (Fig. 1e). The difference in the inhibition between the two mutants may be because of their strategic location of interaction with 16S rRNA. Spectinomycin (SPC), the SSU targeting antibiotic known to inhibit protein synthesis, showed similar inhibition as RafH at 5X concentration (Fig. 1e). Further, the cryo-EM grid preparation conditions were optimized. The cryo-EM image showed an even distribution of intact ribosome particles with optimum ice thickness (Fig. 1f).

### Single particle reconstruction and sorting structural heterogeneity

For elucidating the molecular mechanism of mycobacterial ribosome hibernation, the structure of the 70S ribosome RafH complex was determined by single particle cryo-EM reconstruction using Relion 3.1.4. After initial 3D classification, class 1 (13% particles) showed the presence of E-site tRNA, class 2 (33% particles) showed the presence of RafH, Class 3 (31% particles) appeared to be empty, class 4 (12%) showed the presence of RafH along with E-site tRNA, and 7% particles remained unaligned. Unexpectedly, we found in nearly 25% of total particles selected for 3D classification, a tRNA bound to the E-site of the 70S ribosome, out of which nearly 12% of particles showed both RafH and E-site tRNA bound to the 70S ribosome (Supplementary Fig. 3).

After extensive 3D classification, 153,262 particles were selected from the bound RafH classes. The 3D refinement yielded an initial cryo-EM map of 3.0 Å resolution. Further, the map quality and resolution were improved to 2.8 Å by performing a CTF refinement and particle polishing (Supplementary Fig. 3a). To further improve the density for RafH CTD, these polished particles were subjected to partial signal subtraction from cryo-EM electron density corresponding to the RafH CTD and its interacting partners bS1 r-protein and H54a of 23S rRNA (Supplementary Fig. 3b). A masked 3D classification into five classes, without alignment, was carried out on the subtracted data. Class 1 showed fragmented cryo-EM electron density for RafH CTD, class 2 showed the presence of bS1 protein in addition to RafH CTD, class 3 showed RafH CTD only, class 4 showed RafH CTD, and the remaining 1% in class 5 were unaligned (Supplementary Fig. 3b). The three classes

 

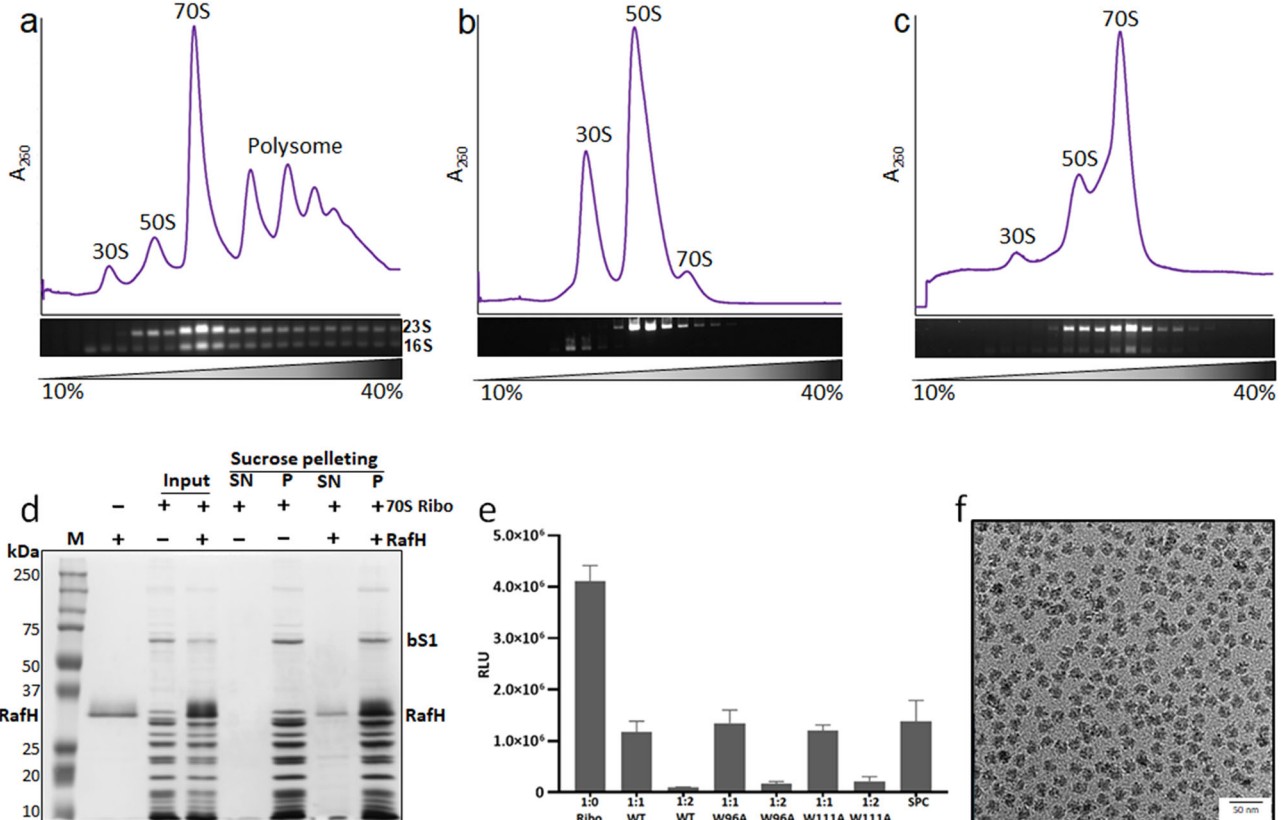

**Fig. 1 | 70S ribosome RafH complex.** The 10–40% sucrose density gradient fractionation profile and corresponding peaks analysis on agarose gel stained with Ethidium bromide(0.2 µg/ml) are shown for (**a**) initial ribosome purification, (**b**) after dissociation and (**c**) after re-association. The 30S, 50S, and 70S are labeled for ribosomal small subunit, large subunit, and associated ribosome, respectively. The 23S and 16S are labeled for rRNA of 30S and 50S, respectively. We obtained the same results for all (>5) ribosome preparation. **d** the 70S ribosomes RafH complex formation and sucrose density pelleting, analyzed on 12% SDS-PAGE, with Coomassie blue staining solution, lane 1 - marker, lane 2 - pure RafH protein, lane 3, 4 - input, lane 5 to 8 – SN (supernatant) and P (pellet) fraction after pelleting on a sucrose cushion. The ribo and bS1 are labeled for ribosome and bacterial ribosomal protein bS1, respectively. **e** In-vitro protein synthesis assay by titrating ribosome and wild type (WT) RafH, W96A RafH mutant, W111A RafH mutant or antibiotic spectinomycin (SPC) at different stoichiometric ratios of 1:1 or 1:2. The RLU (Relative Luminescence Unit) is measured as the production rate of nLuc activity. Data represents as mean ± SEM (standard error mean), where $n = 3$. **f** The 2D cryo-EM micrograph collected during the initial grid screening stage in a JEOL 2200 FS microscope with a Gatan K2 Summit camera. The source data for Fig. 1 is provided in the source data file.

2, 3, and 4, all having cryo-EM electron density for RafH CTD, with a total of 110,934 particles, yielded a 2.8 Å cryo-EM map after 3D refinement and postprocessing (Supplementary Fig. 3b). Multi-body refinement further improved the map quality and resolution to 2.7 Å and 2.9 Å for the LSU and SSU, respectively (Supplementary Figs. 3b, 4 and Supplementary Table 1). A similar approach of multi-body refinement was applied to improve the quality of cryo-EM maps for classes 2 and 4 separately (Supplementary Table 1). The consensus maps of ribosome RafH complexes (map1), with bS1 (map2), and with E-site tRNA (map3) were selected for model building and structure interpretation (Supplementary Fig. 3b).

## Cryo-EM structure of 70S ribosome RafH complex

Overall, the cryo-EM map shows high resolution features (Fig. 2a, b) with distinctly visible secondary structures α-helices and β-sheets for RafH NTD (Fig. 2c). Most of the amino acid residue side chains and nucleotides were clearly visible in our cryo-EM map (Fig. 2c, d, 3 and Supplementary Movies 1–6). The local resolution calculated using ResMap showed the resolution ranges from 2.5 Å to 5.5 Å, with most regions having better than 3.5 Å resolution (Supplementary Fig. 4). Some of the flexible regions, such as RafH CTD, E-site tRNA, L1 stalk, L7/L12 stalk, bS1, uS2, and H54a, having a lower resolution, were interpreted by applying a low pass filter of 3.5 Å resolution to the final maps (Fig. 2a, b and Supplementary Fig. 5). For bS1 r-protein, the two

N-terminus domains, OB1 and OB2, were clearly visible, whereas other parts were disordered (Fig. 2b and Supplementary Fig. 5a, b).

## RafH NTD binds to the conserved binding pocket in the 70S ribosome

RafH is a ~30 kDa protein with 258 amino acid residues. It possesses two domains, the N-terminus domain (NTD), residues 1–100 and C-terminus domain (CTD), residues 131–258. These two domains are connected by a flexible linker region, residues 101–130, which mainly interact with the 16S rRNA of SSU (Figs. 2–4, Supplementary Figs. 6, 7a, 8–10, and Supplementary Table 2). The RafH NTD has a conserved domain having α/β fold with $β_1α_1β_2β_3β_4α_2$ topologies where 4 β-strands form an antiparallel β-sheet and the two α-helices stack to the one side of the β-sheet. A mini helix $α_3$, connects the RafH NTD through a small loop (Figs. 2c, 3, Supplementary Figs. 7a and 9). RafH NTD binds to the cleft between the head and body of the SSU (Figs. 2a, 3 and Supplementary Fig. 10) to a similar binding site reported for HPF[long] NTD, HPF[short,] and YfiA in ribosome structures[10]. At this cleft, the NTD makes extensive interactions with the 16S rRNA, anticodon stem loop of E-site tRNA, r-protein uS9, and also with the inter-subunit bridge B2a (Figs. 2a, 3, Supplementary Fig. 6 and Supplementary Table 2). Some predominant interactions are illustrated (Fig. 3 and Supplementary Movies 1–6). The side chain of residue R75 of the helix α2 interacts with A1477-G1478 of 16S rRNA and A2137 of 23S rRNA (Fig. 3

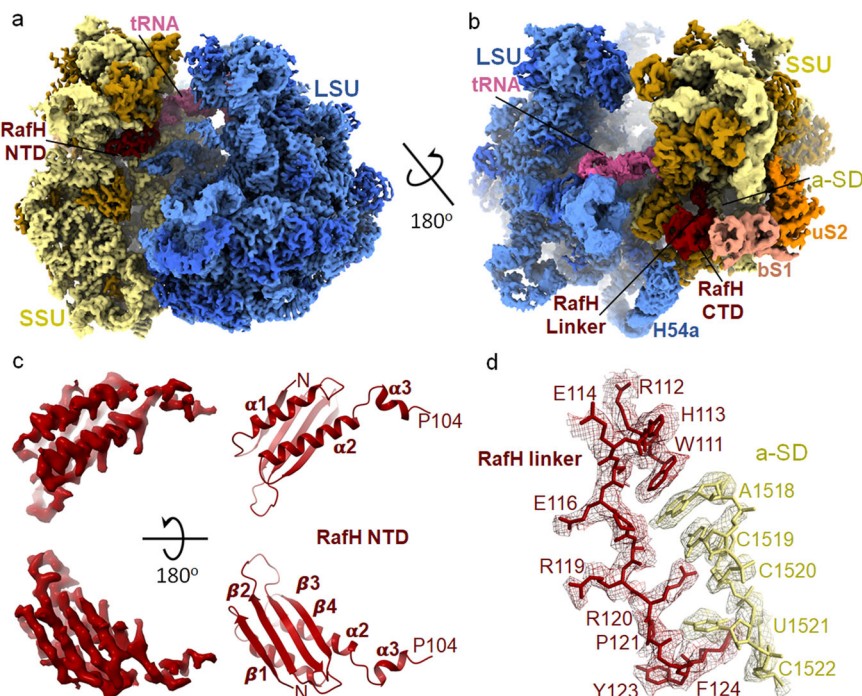

**Fig. 2 | Cryo-EM structure of *Mycobacterium smegmatis* 70S ribosome RafH complex.** The overall architecture of the 70S ribosome RafH complex is shown in the mRNA entry site (**a**) and mRNA exit site (**b**) by a rotation through a diagonal axis. The SSU 16S rRNA (khaki), SSU r-proteins (dark golden), RafH (maroon), tRNA (pink), the LSU 23S rRNA and 5S rRNA (cornflower blue), LSU r-proteins (royal blue), bS1 (dark salmon) and uS2 (orange) are labeled. The single particle reconstruction data processing summary is shown in Supplementary Fig. 3, gold standard FSC and local resolution of final maps are shown in Supplementary Fig. 4, The cryo-

EM maps for individual r-proteins, bS1 and uS2 and E-site tRNA and their model is shown in Supplementary Fig. 5, a full RafH model is shown in Supplementary Fig. 7. **c** The RafH NTD cryo-EM density (left panel) and model in ribbon (right panel), the top panel is rotated by 180° along *X*-axis, and shown in the bottom panel, the secondary structures are labeled. **d** The cryo-EM density in mesh and model in stick style corresponds to RafH linker region residues, 111–124 (maroon), and a-SD (anti-Shine Dalgarno sequence) region of 16S rRNA nucleotides, 1518–1522 (khaki), are shown. For more clarity, an animation is provided in Supplementary Movie 1.

and Supplementary Movie 2), thus providing more stability to 70S ribosome as these nucleotides form an inter-subunit bridge, B2a[49]. The positively charged side chain residues K21, R24, and R28 of the helix α1 make electrostatic interaction with the backbone phosphate of the h44 of 16S rRNA, residues G1478, U1479, C1480, and G1481. The A770 of h24 interacts with the H30 of RafH (Fig. 3 and Supplementary Movie 3). The arginine-rich patch of α2 composed of R84, R88, and R91 interacts with C1382, C1383, and G1384 of the 16S rRNA (Fig. 3 and Supplementary Movie 4). Similarly, the residues R37, R39 of β2 strand, residue Q55 of β3, and R66 of β4 forms a positively charged patch that stacks against U947 and G948 of h31 of the 16S rRNA (Fig. 3 and Supplementary Movie 5). The W96, which harbors in the mini helix α3, makes stacking interaction with the G673 of h23 (Fig. 3 and Supplementary Movie 6). The RafH NTD also interacts with the r-protein uS9 (Supplementary Table 2). The H35 and D59 residues of RafH interact with the C-terminus residue R150 of the uS9 r-protein, whereas the N64 residue of RafH interacts with the K149 of the uS9 r-protein.

## RafH CTD binds to the unique binding site in the 70S ribosome

In the cryo-EM map, the resolution for the RafH CTD was relatively low compared to its NTD (Figs. 2a, b, and 4a, b). However, the RafH CTD model obtained from AlphaFold2 nicely docked in the cryo-EM density designated to RafH CTD (Fig. 4b). We could clearly see α-helices and the side chains for some of the residues; R215, E219, R220, L221, and L223 for one of the α-helices, α5 (Fig. 4b), which has further confirmed its binding site. The RafH CTD binds to the mRNA 'platform binding center (PBC)' composed of proteins bS1, uS7, uS11, and bS18 with 16S rRNA helices h26, h40, and 23S rRNA helix H54a. The uS11 and OB2 domain of the bS1 sandwich the RafH CTD (Figs. 2b and 4a). RafH CTD is composed of two similar protein folds, an α helix with 4 stranded antiparallel β-sheet, having

$β_5α_4β_6β_7β_8α_5β_9β_{10}β_{11}β_{12}$ topologies (Supplementary Figs. 7a and 9). The $β_6β_7β_8β_9$ forms a 4 stranded antiparallel β-sheet where α4 stacks to one side of the β-sheet and form the first protein fold. Similarly, $β_5β_{10}β_{11}β_{12}$ forms another 4 stranded antiparallel β-sheet and α5 stacks to one side of it to form the second protein fold, and a loop region connects the two folds. The β-sheets of each fold stacks nearly parallel to each other and form a dimer like structure[50] (Fig. 4b, Supplementary Figs. 7a and 9). On the contrary, the HPF[long] CTD is composed of a single protein fold (Supplementary Fig. 9) but attains a RafH CTD like architecture by its dimerization (Fig. 4c), as a consequence of which 100S disome[10] formation takes place (Fig. 4b, c).

## The RafH linker interacts with the anti-Shine Dalgarno region of 16S rRNA

A flexible linker connects the RafH NTD and CTD with residues, stretching from 101 to 130 (Supplementary Fig. 7a). The linker residues between W111 to F124 extensively interact with the nucleotide stretch, A1518 to C1522, which harbors the anti-Shine Dalgarno region of the 16S rRNA (Figs. 2b, d, 5, Supplementary Figs. 7, 9, Supplementary Table 2 and Supplementary Movie 1). This remarkable interaction involves residue W111 making a stacking interaction with the A1518 of 16S rRNA. The R120 side chain makes electrostatic interaction with the C1519 base and phosphate of the C1520. The main chain of A118 also interacts with the nitrogenous base of A1518 of 16S rRNA. The main chain of P121 interacts with U1521. The F124 makes a stacking interaction with C1522 of 16S rRNA (Fig. 2d). The linker also interacts with the anticodon stem-loop of the tRNA bound to the E-site of the ribosome (Fig. 2a, b, Supplementary Fig. 11 and Supplementary Table 2). However, in the cryo-EM map, we could not see resolved nucleotides (Supplementary Fig. 5c), maybe the cryo-EM density for E-site tRNA is from averaged tRNAs, as the E-site tRNA co-purified during ribosome purification.

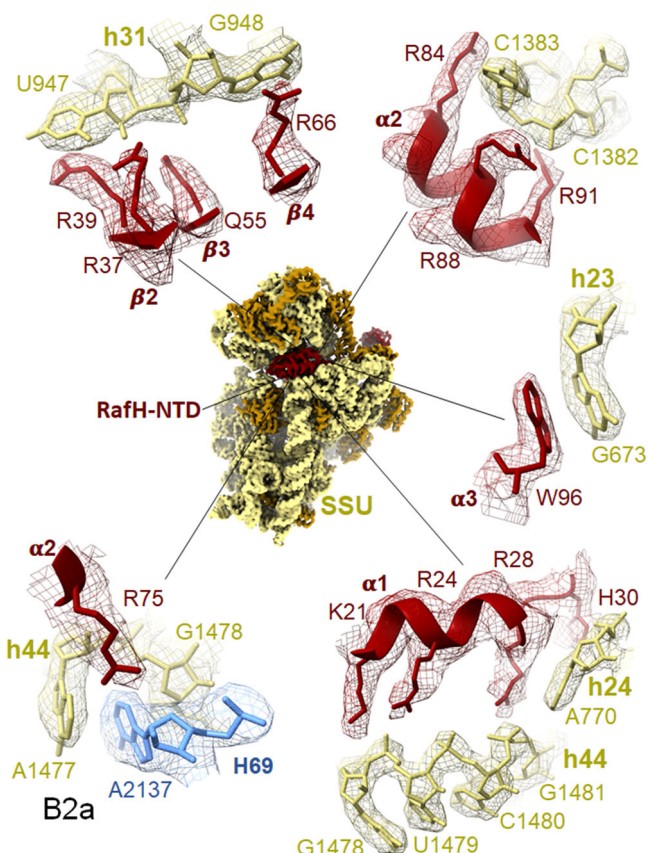

**Fig. 3 | Ribosome and RafH NTD interaction.** The cryo- EM density in surface view for the small subunit with RafH at the center and its magnified regions, where the cryo- EM density in mesh and model in stick and ribbon are shown. For clarity, the ribosomal large subunit is not shown. The RafH 16S rRNA interactions in counter-clockwise, α2 R75 with Bridge B2a (bottom left), α1 with h44 (bottom right), α3 W96 with h23 G673 (middle right), α2 with C1382-C1383 (top right), residues from β2, β3, and β4 with h31 U947, G948 (top left) are shown. For more detail, Supplementary Movies 2–6, Supplementary Fig. 6, and Supplementary Table 2.

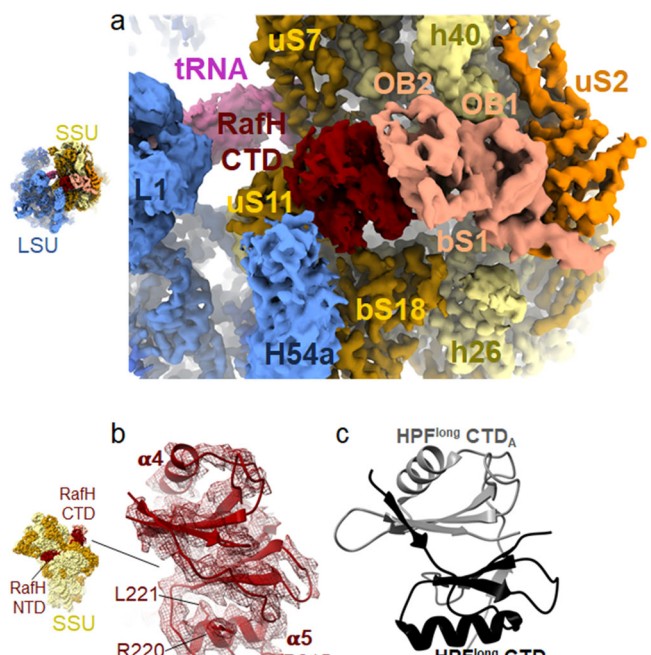

**Fig. 4 | RafH CTD structure and its binding site on the ribosome. a** The RafH CTD binding site present in cryo- EM map in surface style for 70S ribosome RafH complex is shown in the same color scheme used for Fig. 2a, b. A thumbnail for the 70S ribosome is shown on the left. **b** Cryo- EM density corresponding to RafH CTD in mesh, model in ribbon, and stick is shown. The thumbnail is shown on the left. **c** The structure of HPF$^{long}$ CTD dimer (PDB ID; 6T7O) with its first monomer (A) (gray) and second monomer (B) (black) are shown.

## Mycobacterial RafH is a dual domain ribosome hibernation promotion factor

The *M. smegmatis* RafH (Ms_RafH) is a dual domain HPF (Supplementary Fig. 7a). The *M. tuberculosis* RafH (Mt_RafH) structure was predicted through AlphaFold2, which predicts protein domain structures with high accuracy[51]. The Mt_RafH possesses a dual domain like architecture with a similar domain topology to that of Ms_RafH (Supplementary Fig. 7b). The N- and C- terminus individual domains of Ms_RafH and Mt_RafH share high structural similarity (Supplementary Fig. 7c) with root mean square deviation (RMSD) between main chain atoms 0.85 and 0.95, respectively. The Ms_RafH and Mt_RafH share an overall 32% sequence similarity. Mt_RafH has N- and C- terminus extensions of 21and 8 amino acid residues respectively, and 9 amino acid insertion in the loop regions that connects the two C- terminus folds. Whereas the Ms_RafH linker is slightly longer with 10 amino acid residues insertion. However, the N- and C- terminus sequences are more conserved than the loop and terminus regions (Supplementary Fig. 8). Most of the interacting residues are highly conserved, identical, or similar among Ms_RafH and Mt_RafH (Supplementary Fig. 8 and Supplementary Table 2), indicating an analogous functional role.

## Mycobacterial 70S hibernating ribosomes adopt a similar conformation

The small subunit of 70S ribosome in the RafH ribosome structure adopts an unrotated closed conformation in our in-vitro reconstituted 70S ribosome RafH complex (Supplementary Fig. 10a). A comparative analysis was carried out with the earlier reported 30S subunit structure of *M. smegmatis* hibernating ribosome, with P- site tRNA and *M. tuberculosis* ribosome. The small subunit of earlier reported hibernating structures from zinc starved condition ribosome[42] and in the stationary phase ribosome[43], both has copurified MPY factor, and adopts a similar unrotated close conformation (Supplementary Fig. 10a). The superimposition of 30S subunit bound with, RafH (PDB ID; 8WIF), MPY (PDB ID; 6DZK), MPY (PDB ID; 5ZEP), 30S with P- tRNA, and *M. tuberculosis* 30S subunit showed that all structure adopts an unrotated close conformation (Supplementary Fig. 10b) with an RMSD between backbone phosphate atoms 0.8 Å to 1.0 Å among them. Suggesting there may be an insignificant artifact due to in-vitro reconstitution of ribosome RafH complex.

The presence of E-site tRNA in re-associated 70S ribosome was a surprise as we observed 13% of the particles with E-site tRNA alone (class 1) and 12% of particles with E- site tRNA and RafH (class 4) (Supplementary Fig. 3a). Even after the 70S ribosomes were dissociated, the subunits were separated in sucrose gradient with 1 mM MgCl$_2$ concentration and re-associated (Supplementary Fig. 3a). The tRNA bound in E- site to the hibernating ribosomes of the stationary phase has been reported in *M. smegmatis* (Supplementary Fig. 11a)[43]. The tRNA bound to the E-site of 100S ribosome has also been reported[21].

The E- tRNA in RafH 70S ribosome binds to the conserved binding site (Figs. 1a, b, 4a and Supplementary Figs. 3, 11) like that of E- tRNA in stationary phase hibernating 70S ribosome (Supplementary Fig. 11a)[43]. The E- tRNA anticodon stem-loop interacts with the linker regions in both RafH and MPY hibernating 70S ribosome (Supplementary Fig. 11a). In the 70S ribosome RafH structure, E- tRNA binds to the conserved binding site and makes extensive interactions with the 50S subunit (Supplementary Fig. 11b). A similar set of interactions was

earlier reported for tRNA bound to the E-site of *E. coli* 70S ribosome[52]. The A76 nitrogenous base of tRNA CCA end sandwiches between nitrogenous bases, G2645 and C2646 of 23S rRNA, and makes base stacking interactions. Similarly, C75 base stacks with A2656 of 23S

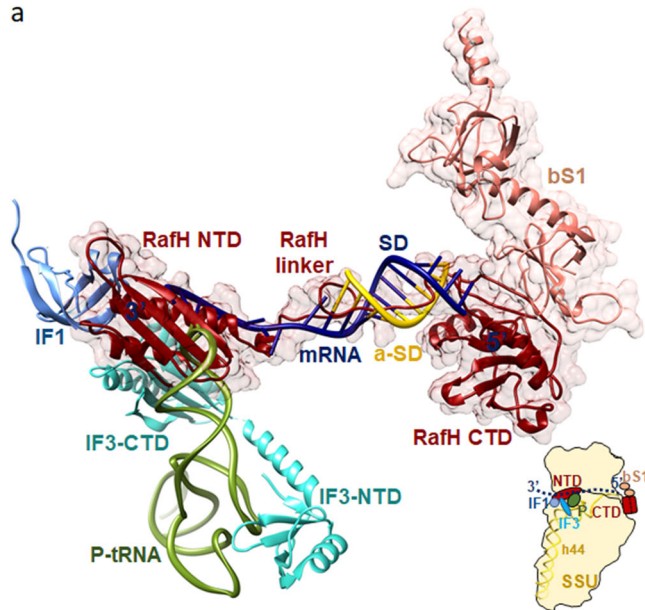

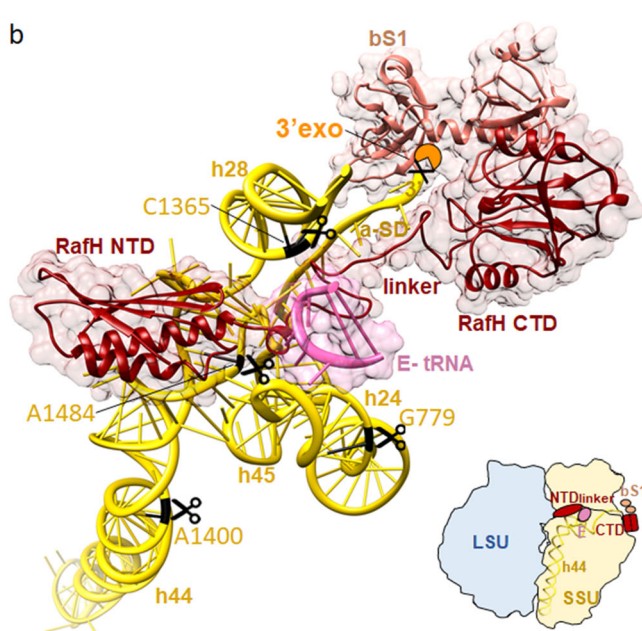

**Fig. 5 | Proposed molecular mechanism for RafH action. a** Inhibition of the translation initiation factor binding by RafH. The pre-translation initiation structure SSU (PDB ID; 5LMT) docked into the ribosome RafH SSU complex structure. For clarity, only the RafH in ribbon (red) with 95% transparent surface, bS1 in ribbon (salmon) with 95% transparent surface, initiation complex factors: mRNA (navy blue), a-SD (gold), IF1 (cornflower blue), IF3 (cyan), and P- tRNA (dark olive green) are shown. A thumbnail is shown in the bottom right. **b** Protection of 16S rRNA from RNase degradation. The RafH (red) and bS1 (salmon) are shown in ribbon with a 95% transparent surface. 16S rRNA helices, h24, h28, h44, h45, aSD, and 3′ of 16S rRNA are shown in a ribbon with a ladder. The E-tRNA anticodon stem loop (hot pink) is shown in a ribbon with a 95% transparent surface. The RNase nucleolytic site predicted in *E. coli* 16S rRNA by ref. 15 and corresponding nucleotides in *M. smegmatis* 16S rRNA are shown in black with the scissors symbol. A thumbnail is shown in the bottom right. The 3′ to 5′ exonuclease RNase PH/RNase R is shown in an orange Pie shape. Its description in 2D is shown in Supplementary Fig. 12.

rRNA (Supplementary Fig. 11). The backbone atoms of G70 and C71 tRNA interact with the backbone atoms of U2068 and U2069 of H68, 23S rRNA. The tRNA elbow region interacts with the L1 stalk of the 50S subunit. Presumably, because of these extensive interactions, the tRNA remains bound in E- site to a small fraction of the 50S subunit even after subunit dissociation at low (1 mM) MgCl$_2$ concentration. Further, we cannot rule out the possibility of a trace of 70S ribosomes in the pool of 50S subunit fraction after sucrose density gradient centrifugation (Fig. 1b).

### RafH would occlude the binding of translation factors, ribonucleases, and antibiotics

To understand the role of RafH in the inhibition of translation, ribosome protection from ribonuclease degradation and antibiotic binding effect, molecular modeling and docking were performed (Fig. 5 and Supplementary Figs. 12, 13). The coordinates of the *E. coli* translation initiation complex (PDB ID; 5LMT)[53] were docked on the 30S ribosome RafH coordinates (Fig. 5a). The RafH NTD binding would overlap with the binding of initiations factors IF1, IF3, P- tRNA and mRNA at decoding site of 30S subunit (Fig. 5a). A similar binding for HPF NTD was reported in earlier hibernating ribosome structures[10]. RafH linker region, which interact with the a-SD of 16S rRNA, would overlap with the binding of the SD sequence of the 5′ UTR of mRNA (Fig. 5a), whereas linker regions of earlier reported structures remain disordered[9]. The RafH CTD, which binds to unique binding site at PBC of the 30S subunit, also engages the bS1 r-protein. Therefore, the bS1 protein would not be available to facilitate the translation initiation in RafH bound form (Fig. 5a).

A recent study showed that in *Δhpf* strain of *E. coli*, the RNA degrading enzymes degraded 16S rRNA by fragmenting at specific sites C764, G799, C1382, G1417, and A1500, then exonuclease further degrades the specific segment[15] (Supplementary Fig. 12). The corresponding sites in *M. smegmatis* are 16S rRNA C744, G779, C1365, A1400 and A1484 (Supplementary Fig. 12), which situates in structurally conserved regions (Fig. 5b). As RafH is known to protect ribosome, particularly its 30S subunit[38]. The RafH and E- tRNA binding would obstruct the binding of RNase to its target sites, G779, C1365, and A1484. (Fig. 5b and Supplementary Fig. 12). Whereas RNase target sites, A1400 and A1484, would not be accessible to RNase in an associated 70S ribosome as these sites are located on the ribosomal interface. Further, the 3′ end 16S rRNA would be blocked by the RafH CTD and bS1 proteins in a hibernating ribosome and probably not accessible to 3′ to 5′ exonuclease RNase PH/RNase R (Fig. 5b and Supplementary Fig. 12). The RNase PH protein from *E. coli* and *M. smegmatis* shares 65% sequence similarity.

The docking of the earlier reported structures of antibiotics bound 30S ribosomal subunit into the 30S RafH complex showed some of the antibiotics would bind in the closed vicinity of the RafH (Supplementary Fig. 13), suggesting a role of RafH in antibiotic resistance.

### RafH CTD and H54a of 23S rRNA would prevent the 100S ribosome formation in Mycobacteria

Molecular modeling and docking were performed to understand the structural basis of RafH's inability to induce ribosome dimerization, resulting in 100S formation. The atomic coordinate of 70S ribosome RafH complex was docked in each 70S monomer of the *Staphylococcus aureus* 100S ribosome dimer (PDB ID; 6FXC). It was found that the RafH CTD binds to a unique position near uS11 r-protein and is surrounded by OB2 of bS1 and H54a of 23S rRNA in mycobacterial 70S ribosome. Besides this, OB1 of bS1 also interacts with the uS2 r-protein (Fig. 6a). But HPF$^{long}$ CTD binds in the same vicinity too, close to the uS2 r-protein (Fig. 6b). Therefore, its binding site overlaps with the binding site of the bS1 r-protein, particularly its OB1 domain in mycobacterial 70S hibernating ribosome (Fig. 6a). As consequences of

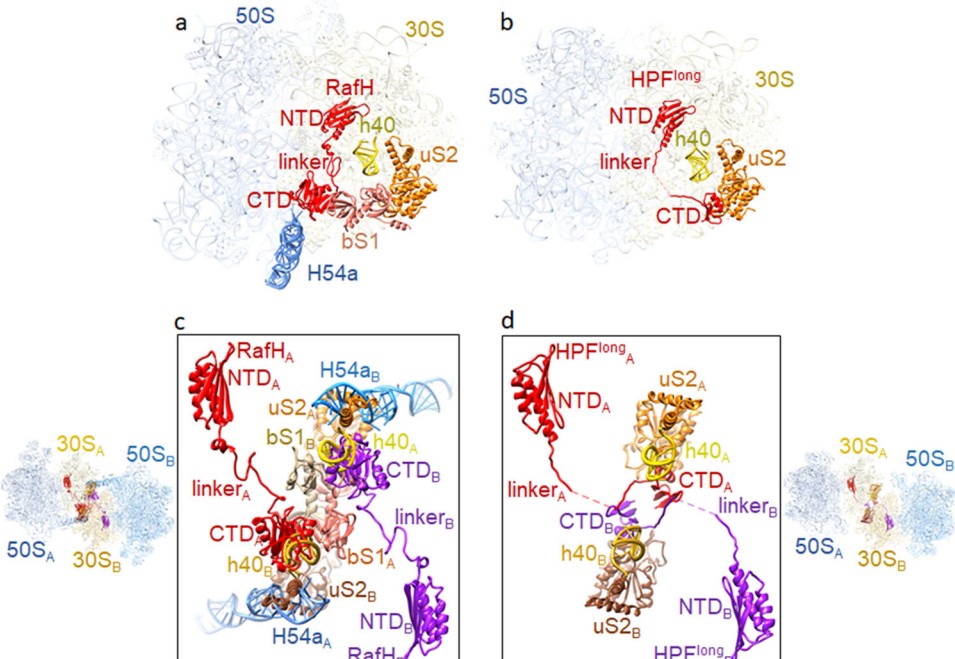

**Fig. 6 | Comparison of RafH binding in 70S ribosome with HPF^long binding in 100S ribosome. a** RafH, bS1, uS2, and h40 of 16S rRNA and H54a of 23S rRNA are shown with LSU and SSU in the 95% transparent background. **b** The corresponding position of HPF^long, uS2, and h40 of 16S rRNA in one of the ribosomes of the *Staphylococcus aureus* 100S structure (PDB ID; 5NGM) is shown with LSU and SSU in the 95% transparent background. **c** Two 70S ribosome RafH complex structures docked into the corresponding positions in *S. aureus* 100S dimer structure (PDB ID; 6FXC) and RafH CTD interacting components are shown in 80% transparent

background on the left side and magnified view with a white background are shown in the box on the right side. One 70S ribosome is labeled as A, and the other 70S ribosome is labeled as B. **d** The HPF^long interacting components uS2 and h40 of 16S rRNA in *S. aureus* 100S ribosome dimer interface (PDB ID; 6FXC) are shown on the right side with 30S and 50S in 80% transparent background, and a magnified view with white background is shown in the box on the left side. Similar to (**c**), one 70S ribosome is labeled as A, and the other 70S ribosome is labeled as B. A multiple sequence alignment among HPFs is shown in Supplementary Fig. 9.

this, *M. smegmatis* 70S (monomer) in an *S. aureus* 100S like dimer architecture would have severe steric clashes at the dimer interface (Fig. 6c) where RafH CTD, H54a and bS1 OB1 of one ribosome would make steric clashes with h40, uS2, and bS1 OB2 of the second ribosome, and vice versa (Fig. 6c). Therefore, would hinder the dimerization of the ribosome. In contrast, the 100S formation is mainly stabilized by HPF^long CTD dimerization. In addition, uS2, bS18, h26, and h40 also stabilize the 100S dimer interface in some species[10] (Fig. 6d).

## Discussion

This structural study reveals a unique mode of mycobacterial ribosome hibernation by RafH, a hypoxia induced HPF. The physiological significance of the RafH has been reported earlier by ref. [38]. RafH being a dual domain HPF, forms hibernating 70S ribosomes. The RafH NTD is conserved and binds to the decoding center of the ribosomal small subunit (Fig. 3). In contrast, RafH CTD is comparatively larger, already having a repeated HPF^long CTD like topology, and forms a similar dimer like architecture as reported for the HPF^long CTD, which is required for 100S ribosome formation (Fig. 4). Therefore, further RafH CTD dimerization is not possible; hence, RafH only forms a hibernating 70S monosome. The H54a and bS1 would also prevent the mycobacterial ribosomes from forming a 100S like architecture (Fig. 6). The RafH binding would block all known critical sites of translation initiation: the decoding center, the a-SD sequence of 16S rRNA, and the bS1 protein at the platform binding center of the ribosomal small subunit. Therefore, RafH inhibits protein synthesis (Figs. 2e, 5a) and would protects ribosome from ribonuclease attack (Fig. 5b), and probably interferes with binding of ribosome targeting antibiotics (Supplementary Fig. 13). Thus, RafH has a distinct mode of ribosome hibernation (Figs. 5 and 7).

The RafH NTD possesses a conserved structural fold and binds to a similar binding site to that of HPF^long NTD, HPF^short, and YfiA[9,10].

RafH NTD interacts with SSU predominantly through electrostatic interactions. However, the additional interaction of RafH R75 residue with the inter-subunit bridge B2a (Fig. 3 and Supplementary Movie 2) suggests that the RafH binding further stabilizes the inter-subunit interaction. The bridge B2a is also known to be involved in the translation initiation and translational processivity in addition to strengthening the inter-subunit interaction[54]. The RafH NTD binding would impede the binding of translation initiation factors (Fig. 5a). Besides this, the RafH NTD binding would also explicitly block recruitment of leaderless (without 5′ UTR) mRNA[55] onto SSU and consequently blocking its translation initiation too. It is a notable observation because in Mtb, nearly 25% of mRNAs are the leaderless mRNA[56], and it is believed that Mtb switches to the leaderless mRNA translation over leadered (with 5′ UTR) mRNA, as a survival strategy under stress[57]. The 30S subunit decoding center is also known as the target sites for antibiotics, particularly the aminoglycoside class of antibiotics, (Supplementary Fig. 13). Therefore, the RafH binding would occlude the binding of these aminoglycoside class of antibiotics, probably through a similar mechanism observed earlier for the aminoglycoside resistance in Mtb[42] and in *Listeria monocytogenes*[58].

The linker region, which connects the two domains, is of varying length and found to be disordered in the HPF^long structures reported so far[10], with this, it was also propounded that the HPF^long linker might not interact with the a-SD sequence of 16S rRNA[22]. Noteworthy, we found that in mycobacteria, the RafH linker regions interact with a-SD sequence of 16S rRNA through electrostatic and base stacking interactions (Fig. 2d and Supplementary Movie 1). This binding would block the interaction between mRNA SD sequence and 16S rRNA a-SD sequence (Fig. 5a). This is critical for correctly positioning the mRNA in the 30S subunit during translation initiation[13]. Thus, RafH presence

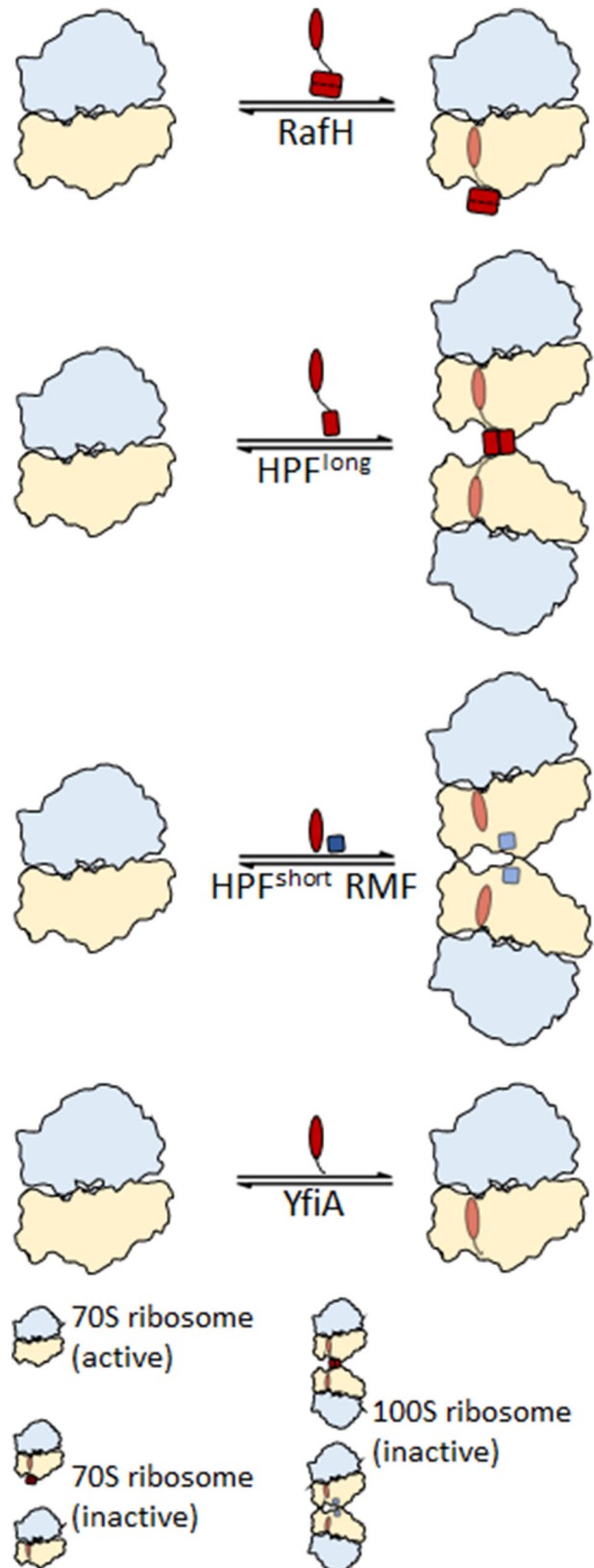

**Fig. 7 | Different modes of ribosome hibernation.** A schematic presentation for the different modes of ribosome hibernation. Top, RafH mediated hibernation in 70S form (from this study). Second from top, HPF[long] induces ribosome dimerization and formation of 100S disome[18–22]. Third, from the top, HPF[short] and RMF[23,24] induce ribosome dimerization and 100S ribosome formation. Bottom, YfiA hibernates ribosome in the 70S form[26,27].

would also block the translation initiation of leadered (with 5′ UTR) mRNA.

The RafH CTD, which is longer than HPF[long] CTD, binds at the PBC and is sandwiched between bS1 and uS11 r-proteins. The PBC has been proposed as the binding site of mRNA before translation initiation and regulates the initiation process[59]. Therefore, the presence of RafH CTD would block its binding (Fig. 5a). Previously reported structures for HPF[short]/RMF and YifA showed bS1 involvement in ribosome hibernation[24,27]. However, no such contribution is described for HPF[long] mediated 100S ribosome formation. Interestingly, we found that bS1 r-protein was present in a small fraction (23%) of the total particles (Supplementary Fig. 3b), suggesting its involvement in RafH mediated ribosome hibernation and further stabilizing the RafH CTD binding. RafH binding would also block the bS1 r-protein mediated translation initiation. The bS1 r-protein is known to be involved in the recruitment of the mRNA's, having AU-rich sequence (weak SD sequence) elements instead of AG-rich SD sequence (strong SD sequence) elements in its 5′ UTR region[60].

The overlapping between the RafH NTD binding site with the HPF[short] suggested that the RafH would protect the 16S rRNA most likely in a similar way (Fig. 5b) as reported in *E. coli*. As the 3′ end of 16S rRNA is blocked by the RafH CTD and bS1 r-protein, it further suggests that in hibernating ribosomes, the 3′ end would not be accessible to 3′ to 5′ exonucleases, RNase PH/RNase R (Fig. 5b and Supplementary Fig. 12). RafH functional insights, particularly the ribosome protection from degradation, have been gleaned by comparison with the literature mostly available in *E. coli*, a widely used prokaryotic model organism. However, it remains to see how RafH protects ribosome degradation in the pathogenic *M. tuberculosis* during hypoxia.

Additionally, the docking studies revealed that the bS1 r-protein binding site in the mycobacterial 70S ribosome (Fig. 6a) overlaps with the HPF[long] CTD binding in 100S ribosome (Fig. 6b) thus, bS1 in mycobacteria would prevent the formation of 100S like ribosome complex, in addition to the severe steric hindrance caused by RafH CTD and H54a at the dimer interface (Fig. 6c). The H54a of 23S rRNA, unique to the mycobacterial ribosome[39,41] adopts a different conformation and appears to interact with RafH CTD (Figs. 2b and 4a). This interaction would further strengthen the 70S stability and also suggests, a role of H54a in ribosome hibernation. A similar conformation was observed in earlier reports in stationary phase hibernating 70S ribosome[43].

The E- site tRNA in 70S ribosome was unexpected as the ribosomal subunits were separated in low MgCl$_2$ concentration before re-association to 70S ribosome. The E- site tRNA has been reported earlier in *M. smegmatis* 70S hibernating ribosomes[43] and *Staphylococcus aureus* 100S hibernating ribosome[21]. The presence of E-site tRNA in 70S with and without RafH (Supplementary Fig. 3) indicates that E-site tRNA binding might not influence the RafH binding. Maybe E-site tRNA binding further stabilizes the associated 70S ribosome and protects it from RNase degradation (Fig. 5b).

The MPY NTD, another HPF known to mycobacteria, binds to the similar RafH NTD binding site, whereas the linker region and CTD remain disordered in reported cryo- EM single particle reconstruction maps[42,43]. Interestingly, MPY possesses amino acid residues similar to HPF[long], which is shorter than the RafH (Supplementary Fig. 9). Likewise, HPF[long] CTD, the two MPY CTD could dimerize and consequently induce ribosome dimerization and 100S formation. Nevertheless, MPY hibernates ribosomes in the 70S form only[38,42,43], and its CTD structure remains unknown[42,43]. It would be interesting to see what prevents MPY CTD dimerization once the binding site in the 70S ribosome is resolved. Perhaps, the H54a and bS1 would prevent the two ribosomes from coming close to forming 100S like ribosome dimers. However, it needs further experimental validation.

*M. smegmatis* has been widely used as a model system to study tuberculosis because of its non-pathogenicity and easy handling[61–64] and similarity with pathogenic *M. tuberculosis* in many aspects[65], especially with respect to the process of protein synthesis[44,66,67]. Because of conserved mycobacterial 70S ribosome architecture and translational process[40–44,67] and structural and sequence similarities of RafH proteins (Supplementary Figs. 7, 8), *M. smegmatis* is, arguably, a suitable model for understating ribosome hibernation in dormant *M. tuberculosis*. Albeit, it would be interesting to get direct functional insights into ribosome stabilization in an actual pathogen, *M. tuberculosis*, during latent tuberculosis infection.

Nevertheless, the RafH is an actinobacteria specific HPF. Thus, mycobacteria have evolved with distinctive RafH mediated ribosome hibernation exhibiting a noble way of translation inhibition, antibiotic resistance, and stabilizing the 70S structure. Therefore, the structure-based design of the modulator of mycobacterial ribosome hibernation may offer a promising strategy to prevent Mtb's entry into the LTBI and shorten the length of TB treatment with a reduced chance of disease relapse.

## Methods

### Ribosome isolation and purification

The ribosomes from *M. smegmatis* (MC$^2$155) were isolated following a similar protocol as reported earlier[68]. The cells were grown at 37 °C till the mid-log phase (0.6 $OD_{600}$) in Sauton's media and pelleted at $7500 \times g$ for 30 min. The cells were lysed using Mixture Mill MM500 (Retsch) for six cycles, each at 30 hertz for 1 min in cryo-condition. A lysis buffer (20 mM HEPES pH 7.4, 20 mM $MgCl_2$, 100 mM $NH_4Cl$, 1 mM PMSF, 3 mM DTT, 1X Protease inhibitor cocktail, for details see Supplementary Note 1) was used to resuspend the cell lysate. Cell debris was removed by centrifugation at $20,000 \times g$ for 30 min. The clear supernatant was layered on a 1.1 M sucrose cushion in buffer A (20 mM HEPES pH 7.4, 20 mM $MgCl_2$, 100 mM $NH_4Cl$, 3 mM DTT) in 1:1 ratio and ultracentrifuged at $100,000 \times g$ using rotor P70AT2 (Hitachi). The crude ribosome pellet was dissolved in buffer B (20 mM HEPES pH 7.4, 20 mM $MgCl_2$, 50 mM $NH_4Cl$, 3 mM DTT) and homogenized using a Dounce homogenizer followed by DNaseI treatment (3 U/µl, ThermoFisher) for 1 h on ice. Subsequently, it was centrifuged at $20,000 \times g$ for 30 min at 4 °C. The concentration of the crude ribosome in the supernatant was estimated by measuring absorbance at 260 nm. For further purification, 10–15 O.D units of crude ribosomes were layered on a 10–40% sucrose gradient in buffer C (20 mM HEPES pH 7.4, 20 mM $MgCl_2$, 30 mM $NH_4Cl$, 3 mM DTT). The gradients were prepared by using BioComp Gradient Master and then ultracentrifuged at $256,400 \times g$ for 4.5 h (P40ST rotor Hitachi) and fractionated using a Gilson fractionator in a BioComp station (Fig. 1a). The fractions from the sucrose gradient fractionation were analyzed, without prior phenol extraction, on 2% agarose gel and 0.06% bleach stained with Ethidium bromide (0.2 µg/ml) (Fig. 1a). The 30S, 50S, 70S, and polysome fractions were concentrated separately using 100 kDa Amicon (Millipore) and stored in buffer D (20 mM HEPES pH 7.4, 20 mM $MgCl_2$, 30 mM $NH_4Cl$, 3 mM DTT).

### Ribosome dissociation and re-association

For dissociation of the 70S ribosome to respective subunits, 30S and 50S, the $MgCl_2$ concentration was reduced from 20 mM to 1 mM by passing 10 ml buffer E (20 mM HEPES pH 7.4, 1 mM $MgCl_2$, 30 mM $NH_4Cl$, 3 mM DTT, 0.1 mM spermidine) three time followed by incubation on ice for 3–4 h, and finally concentrated using 100 kDa Amicon (Millipore). The ribosomes were layered on a 10–40% sucrose gradient prepared in buffer E, and ultracentrifugation was carried out at $256,400 \times g$ for 4.5 h (P40ST, Hitachi). The gradients were fractionated and again analyzed on a 2% agarose gel with 0.06% bleach and stained with 0.2 µg/ml Ethidium bromide (Fig. 1b). The peaks corresponding to 50S and 30S were concentrated separately, and concentration was estimated by measuring absorbance at 260 nm. The 70S ribosomes were re-associated by mixing the equimolar concentration of the 50S and 30S ribosomes, and the concentration of $MgCl_2$ was increased from 1 mM to 20 mM. These re-associated ribosomes were analyzed similarly by density gradient ultra-centrifugation (Fig. 1c).

### RafH overexpression and purification

The *M. smegmatis* gene MSMEG_3935 encoding RafH protein was commercially synthesized from GenScript and cloned in the pET-28a (+) bacterial expression vector with C-terminal containing His$_6$-tag. The presence of MSMEG_3935 was confirmed by double digestion using NdeI and Xho1 restriction enzymes. The RafH protein was overexpressed in *E. coli* C41(DE3) cells. The cells were grown at 37 °C till 0.6 $OD_{600}$, cell culture was chilled at 4 °C for 30 min, and then RafH overexpression was induced by adding 0.5 mM IPTG. The cells were further grown at 16 °C for 16 h at 180 rpm and then pelleted at $7500 \times g$ for 30 min. The cells pellet was lysed by sonication in lysis buffer (50 mM Tris-HCl pH 7.0, 500 mM $NH_4Cl$, 10% glycerol, 20 mM Imidazole, 0.5% Tween 20, 10 mM $MgCl_2$, 1 mM PMSF, 1x Protease inhibitor cocktail (cOmplete, EDTA- free tablets (Roche) and 5 mM β-ME). Then, the lysate was pelleted down at $20,000 \times g$ for 1 h at 4 °C. The supernatant was incubated with Ni-NTA (Millipore) beads for 2–3 h on a rocking shaker at 4 °C followed by 3 times washing with wash buffer (50 mM Tris-HCl pH 7.0, 500 mM $NH_4Cl$, 10% glycerol, 20 mM Imidazole, 10 mM $MgCl_2$, 5 mM β-ME) to remove the non-specific bound protein. Then, the protein was eluted (1 ml fraction) with 20 ml elution buffer (50 mM Tris-HCl pH 7.0, 500 mM $NH_4Cl$, 10% glycerol, 300 mM Imidazole, 10 mM $MgCl_2$, 5 mM β-ME). The protein fractions were pooled and concentrated in a 10 kDa cut-off Amicon filter (Millipore). Further protein purification was performed by size-exclusion chromatography using Superdex$^{Tm}$ 200 increase 10/300 column (Cytiva). The protein purity was confirmed with 12% SDS-PAGE (Fig. 1d). The RafH protein fraction was pooled and concentrated in a 10 kDa cut-off Amicon filter (Millipore). The protein concentration was checked by measuring absorbance at 280 nm, and protein at 1.2 mg/ml concentration was stored at −80 °C.

The RafH point mutants, W96A and W111A, were generated by site-directed mutagenesis (see Supplementary Note 2). Both residues, W96 and W111, make base stacking interactions with the 16S rRNA nucleotides, G673 and A1518, respectively (Figs. 2d and 3). To generate RafH W96A mutant, the forward primer W96A 5′ TATTGCGGAGCACGCGGAAGCGCGTCG 3′ and the reverse primer W96A- 5′ CGACGCGCTTCCGCGTGCTCCGCAATA 3′ primers were used. Similarly, to generate RafH W111A mutant, the forward primer W111A 5′ GCGGGTCGTGAAGCGCGTCATGAGAGC 3′ and the reverse primer W111A- 5′ GCTCTCATGACGCGCTTCACGACCCGC 3′ were used. The primers were commercially synthesized from G-Biosciences. A Phusion® High-Fidelity DNA Polymerase kit (NEB) was used with RafH wild-type plasmid pET28a (+) as a template for PCR amplification. A standard protocol for PCR reaction was performed. 10 units of DpnI (NEB) enzyme was added to digest parental plasmid of PCR product and incubated at 37 °C for 1 h. 20 µl of Dpn1 digested reaction was transformed to DH5α (ultra-competent cells) *E. coli* cells. The plasmids were isolated from transformed cells. The presence of mutation was confirmed using DNA sequencing. The RafH mutants were purified by following a similar protocol as wild-type RafH protein.

### Ribosome RafH complex preparation and sucrose pelleting assay

The ribosome RafH complex was prepared in 100 µl reaction volume by incubating 1 µM 30S with 1 µM 50S at 37 °C for 10 min in a complex-binding buffer (20 mM HEPES pH7.4, 20 mM $MgCl_2$, 100 mM $NH_4Cl$, 3 mM DTT) followed by incubation on ice for 5 min. 10 µM of RafH protein was added to this reaction mixture, and 10 µl of buffer was added to the control sample and incubated for 20 min at 37 °C. 80 µl of

this reaction mixture was layered on a 0.8 M (500 μl) sucrose cushion in a 1 ml open thick wall polypropylene tube. The complex was pelleted down by ultracentrifugation at 600,000 × *g* for 4 h in a Beckman Coulter rotor MLA-150. The pellet was resuspended in a 50 μl complex binding buffer, and supernatant was concentrated using 10 kDa cut-off Amicon (Millipore) till volume reached 50 μl. Further, the presence of RafH was investigated by running supernatant and pellet fractions of both reaction and control samples on 12% SDS-PAGE stained with Coomassie blue staining solution (Fig. 1d).

### In-vitro translation inhibition assay
A luminescence based translation inhibition assay was performed using an in-vitro translation PURExpress® Δ Ribosome Kit from (NEB) with the pMSR plasmid, having the nLuc gene. The constituents of the kit were incubated with 50 ng/μl pMSR DNA template, 1U μl$^{-1}$ murine ribonuclease inhibitor (thermo scientific), 2.4 μM crude ribosomes, with 1X and 2X molar higher concentration of RafH (wild type) or its mutants, W96A or W111A, and 5X Spectinomycin (at 5X it shows a similar inhibition as RafH protein), at 37 °C for 3 h. A total of eight reactions in triplicates with 10 μl reaction volume were incubated, then the reaction was quenched by keeping the reaction mixture on ice for 10 min. The luminescence, relative luminescence unit (RLU), was measured immediately after adding 30 μl NanoGlo substrate by using a GLOMAX luminometer from Agilent Technology (Promega). The data was plotted using GraphPad prism 8.0.1. (Fig.1e).

### Electron microscopy
For preliminary screening, negative staining was performed. 3 μl of 1 mg/ml 70S ribosome RafH complex was applied on a glow discharged 300 CF300-Cu grids (EMS). The excess sample was blotted, washed with MilliQ water, and stained with 1% uranyl acetate solution. The grids were screened in JEOL 1400 JEM, 120 kVa microscope. The cryo-EM grids were prepared using Gatan's CP3 plunger for cryo-EM condition optimization. 3 μl sample was applied on a glow discharged grid R 1.2/1.3 on 300 mesh Cu Quantifoil from TED PELLA, INC and blotted for 3 s before plunging grids into the liquid ethane. The grids were mounted on Gatan 626 Cryo-holder and analyzed in JEOL 2200 FS JEM, 200 kVa microscope equipped with the Gatan K2 Summit direct electron detector camera. The data was collected at a low dose of 1.3 e/Å$^2$/frame in movie mode, 30 frames per movie stack at 1.3 Å pixel size by using JEOL's automatic data collection software, JADAS (Fig. 1f). All the initial sample optimization and grid screening were done at the Advanced Technology Platform Center (ATPC), Regional Center for Biotechnology (RCB), Faridabad.

The high-resolution data was collected using 300 kVa Titan Krios (ThermoFisher) equipped with Falcon 3 direct electron detector camera at National Electron Cryo-Microscopy Facility, Bangalore Life Science Cluster (BLiSc), Bangalore. The data was collected in an electron counted movie mode. 12,343 movie stacks were collected with 25 movie frames per stack at 1.07 Å pixel with an electron dose of 1.34 e/Å$^2$/frame (Supplementary Fig. 3a and Supplementary Table 1).

### Single particle reconstruction
The single particle reconstruction was carried out using Relion 3.1.4[69]. A summary of data processing is given in Supplementary Fig. 3 and Supplementary Table 1. The movie frames were drift corrected, and single micrographs were generated using Relion 3.1.4. The micrographs were CTF corrected using CTFFIND4[70]. 1,202,461 auto-picked particles were subjected to two rounds of 2D classification, and the best 2D classes containing 730,969 particles were selected. These particles were subjected to 3D classification, and classes showing density for RafH, containing 328,619 particles, were subjected to 3D refinement. A 60 Å lowpass filtered 70S ribosome cryo-EM map (EMDB ID; 8932)[42] was used as a reference map. A focused 3D classification on a small subunit without alignment was performed. The one class which

shows apparent density for RafH CTD with 153,262 (47%) particles yielded a cryo-EM map at 3.0 Å resolution after 3D refinement and postprocessing. The gold-standard FSC = 0.143 criterion[71] was used for resolution estimation. The CTF refinement and particle polishing were used to further improve the resolution to 2.8 Å (Supplementary Fig. 3a).

However, the density for the RafH CTD was weaker than expected for a map at this resolution. Therefore, a focused 3D classification was performed with signal subtraction without alignment (FCwSS) with a regularization parameter of T = 12[72]. A partial signal subtraction of cryo-EM electron density corresponding to the RafH CTD and its interacting partners bS1, H54a from polished particles were carried out, and FCwSS was performed by classifying into 5 classes (Supplementary Fig. 3b). Class 1, with 44,299 (28%) particles, showed fragmented cryo-EM density. Class 2, with 36,121 (23%) particles, contains RafH CTD and bS1. Class 3, with 30,514 (20%) particles, contains only RafH CTD. Class 4, with 44,299 (28%) particles, contains RafH CTD and E-site tRNA. Class 5 contains 1% unaligned particles. Classes 2, 3, and 4 were subjected separately for 3D refinement and postprocessing. Final maps were interpreted by applying a 3.5 Å resolution low pass filter as RafH CTD still has a lower density than the ribosome core. The particles from these three classes, 2, 3, and 4, were joined together with a total of 110,934 particles, which yielded a final map of 2.8 Å resolution, upon 3D refinement and postprocessing (Supplementary Fig. 3b). The local resolution for the final maps was calculated using ResMap[73] (Supplementary Fig. 4).

To deal with the inherent ribosomal inter-subunit motion, a 3D multi-body refinement[74] was carried out by treating LSU and SSU as two bodies with 10 Å rotation and 2-pixel translation. It has further improved the map quality of individual subunits and yielded the final resolution of 2.7 Å and 2.9 Å for LSU and SSU, respectively (Supplementary Fig. 3). Similarly, a multi-body refinement was carried out for class 2, which contains bS1, and class 4 which has E-tRNA in addition to RafH. The final maps were low pass filtered to 3.5 Å resolution (Supplementary Fig. 3b) because of poor resolution for CTD, bS1, and E-tRNA compared to the core of the ribosome.

### Model building and structure analysis
The atomic coordinates of *M. smegmatis* 70S ribosome (PDB ID; 6DZI)[42] was docked in the final cryo-EM map using Chimera[75]. The refinement was performed using phenix.real_space_refinement[76]. The model building was carried out using COOT v.0.9.3[77]. For RafH and bS1 model building, the initial models were obtained from AlphaFold2[51] and docked in cryo-EM map. The linker region was manually built in COOT. The final model quality was checked using MolProbity[78]. Figures were prepared in Chimera and ChimeraX[79].

### Reporting summary
Further information on research design is available in the Nature Portfolio Reporting Summary linked to this article.

## Data availability
Nine cryo-EM maps have been deposited in the EMDB (https://www.ebi.ac.uk/emdb/) and the atomic coordinates have been deposited in the wwPDB (https://www.wwpdb.org) with accession codes: EMDB-37551 and 8WHX for 70S ribosome and RafH, EMDB-37552 and 8WHY for 50S (body 1 of 70S ribosome and RafH), EMDB-37565 and 8WIF for 30S (body 2 of 70S ribosome and RafH), EMDB-37559 and 8WI7 for 70S ribosome RafH and bS1, EMDB-37560 and 8WI8 for 50S (body 1 of 70S ribosome RafH and bS1), EMDB-37561 and 8WI9 for 30S (body 2 of 70S ribosome RafH and bS1), EMDB-37562 and 8WIB for70S ribosome RafH and tRNA, EMDB-37563 and 8WIC for 50S (body 1 of 70S ribosome RafH and tRNA), and EMDB-37564 and 8WID 30S (body 2 of 70S ribosome RafH and tRNA). Source data are available as Source Data file. Source data are provided with this paper.

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

## Acknowledgements

We acknowledge the Electron Microscopy facility at the Advanced Technology Platform Center, Regional Center for Biotechnology, Faridabad, and thank Dr. Reena and Mr. Madhava for help in initial cryo- EM sample screening. We acknowledge the National Electron Cryo- Microscopy Facility, Bangalore Life Science Cluster (BLiSc), Bangalore, and special thanks to Drs. Vinothkumar Kutti and Sucharita Boss for cryo- EM data collection. We thank Dr. Todd Gray, Wodsworth Center Albany, USA, for providing the plasmid used in the in-vitro translation assay. N.K. and S.S. acknowledge fellowship from UGC and DBT, respectively. This work is supported by an Early Career Research Award grant, ECR/2018/001944, to P.S.K. from the SERB Department of Science & Technology, India.

## Author contributions

P.S.K. conceived this study. N.K. purified the RafH protein and created its mutants. S.S. purified the ribosome. N.K. prepared the ribosome RafH complex with the help of S.S., N.K., and S.S. performed the in-vitro translation assay. P.S.K. and N.K. collected initial data. P.S.K. and N. K. performed the single particle reconstruction. P.S.K. prepared the figures and wrote the manuscript with the inputs for N.K. and S.S.

## Competing interests

The authors declare no competing interests.
