## [Peer Review File · Nature Communications]

Cryo- EM structure of the mycobacterial 70S ribosome in complex with ribosome hibernation promotion factor RafHREVIEWER COMMENTS

Reviewer #1 (Remarks to the Author):

The article by Kumar et al., reports the first cryo-electron microscopy (cryo-EM) structure of *Mycobacterium smegmatis* RafH hibernation promotion factor (HPF) bound to the ribosome at a near-atomic resolution. Overall, and despite a manuscript carelessly edited, the experiments are handled with care and the results are original. They notably explain why this specific HPF triggers ribosome hibernation of 70S monosomes and not 100S disomes.

Major comments:

- The introduction is too long and would benefit from a shortening in order to focus more on hibernation in *Mycobacteria*. Covid and epidemics are certainly out of scope for this structural work. Moreover all the results are described with a lot of details at the end of the abstract and of the introduction. This weakens the following chapters of the manuscript

- Results: the authors dissociate and reassociate 70S ribosomes in order to “remove the co-purified P-site bound tRNA”. This step is indeed important to remove tRNAs, but also mRNA, translation factors etc. from the ribosomes. Therefore, why do the authors focus on P-site tRNAs? Indeed, they then observe (see below) tRNAs in the E-site and try to interpret this presence as a strategy to hoard tRNAs (page 13). The fact is that the authors use a high MgCl₂ concentration (20mM) that could trap E-site tRNAs that were also supposed to be removed. Please comment.

- How do you explain the presence of E-Site tRNA after dissociation / purification and reassociation and is it compatible with RafH activity model ? What about an mRNA into the ribosomal path?

- The authors use RafH protein from *M. smegmatis* in order to keep the complex with 70S homologous. However, a comparison (sequences, 3D models) with *M. tuberculosis* RafH would be useful.

Minor comments

Fig 1f has no scale bar

Fig6 is too big for such a simple sketch.

Page 7, top: « signal subtraction »: of what?

Page 7: “cryo- EM mass”: replace by “mass” by “electron density”

Page 7: "These 3 maps, of ribosome RafH complexes". Use rather "consensus map"

Page 9: “The signal for RafH CTD and its surrounding regions, H54a, bS1, and uS2, was subtracted”. It is the contrary, isn't it?

Reviewer #2 (Remarks to the Author):

The manuscript "Cryo- EM structure of the mycobacterial 70S ribosome in complex with ribosome hibernation promotion factor RafH, reveals the unique mode of mycobacterial ribosome hibernation" by Niraj Kumara, Shivani Sharma & Prem S. Kaushala reports on high resolution molecular structures of in vitro formed complexes between ribosome subunits purified from *Mycobacterium smegmatis* and His-tagged RafH expressed in, and purified from, *Escherichia coli*. RafH is a hibernation promotion factor expressed during hypoxia as a member of the DosR regulon in *Mycobacterium*. A structure of this complex has not been reported earlier and the CTD of RafH was found to bind to a hitherto unique position (PBC) at the 30S subunit.

I have to state that I am not an expert in cryo-EM and that I am unable to evaluate the technical details of these experiments reported here. So to that part I can only say that the structure figures are clear and well explained.

I have two major concerns with this manuscript

The first two paragraphs of this manuscript contains facts about tuberculosis that is, in my view, unrelated to the scientific part of the work presented. Clearly, motivation and funding is important but I think this is overwhelming. Further, along the same road, I also think the use of Mtb as an alias for *M. smegmatis* (e. g. p12) is a little dishonest, because Mtb means *M. tuberculosis* in my dictionary. In general you try to broaden your conclusions to all mycobacteria but why is *M. tuberculosis* a pathogen

and *M. smegmatis* not? There must be differences and if I were mean I could suggest it is because *M. tuberculosis* is better at coping with the hypoxia in macrophages.

I would have used some of the space in the introduction to make a figure (maybe like Fig.S6 only highlighting homology) comparing known HPFs found at ribosomes of *M. smegmatis* to HPFs found in *Mtb* and then at various conditions. I think that would make the paper more readable for the general reader and maybe point clearer to the new findings and conclusions of this paper.

The second major concern is: has the in vitro formed complexes used here anything to do with in vivo complexes? I understand that the uniform formation of a high number of identical complexes promotes a high-resolution cryo-analysis. However, I begin to worry when I learn that some of the ribosomal proteins are specific for the state of growth (or stress) in *M. smegmatis* like Zn depletion (your ref. 31; doi/10.1073/pnas.1804555115). In the present work you are purifying ribosomal subunits from a mid-log phase growing culture and combine them with a RafH protein, His-tagged and purified from *E. coli*. This is somehow fair, but you should at least discuss the possible pitfalls in this methodology when you try to broaden your conclusions as far as you do.

In your reference 37 (DOI 10.1074/jbc.M112.364851) they purify ribosomes from hypoxia stressed *M. smegmatis* cultures and find along RafH also MSMEG_1878 (alias MPY) bound to the ribosomes, but since it is an ensemble analysis no details are available if the factors bound to the same ribosomes together or if they bound alone to different ribosomes. Is it possible that the binding you see could be, at least partly, missing important interactions/information due to the preparation method and use of purified mid-log growth phase ribosome subunits?

Minors

p4 last paragraph) RafH is described to be overexpressed during hypoxia stress as a member of the DosR regulon. Overexpression is a strange word. I guess you mean that expression is increased?

p13) Doesn't inhibition of translation by blocking the anti-SD demand that RafH can bind 30S subunits alone? I assume that the 30S containing pre-initiation complex is expected to initiate translation on SD-bearing mRNAs by pairing between SD and anti-SD. Later the 50S subunit joins and translation begins. Is there any evidence of RafH being able to bind to 30S alone?

Instead of this rather hypothetical intervention of translation initiation I think it could be worth considering if, and how, RafH could stabilize rRNA by its binding. It could probably be the most prominent function of RafH in vivo. In (DOI 10.1074/jbc.M112.364851) they find very unstable rRNA in a

Delta(dosR) strain of which the phenotypes are reverted when ectopic rafH is expressed from a plasmid. A hint could be that in Escherichia coli, the degradation of rRNA begins when the subunits are free, with a RNasePH attack in 16S RNA in the anti-SD end. (Basturea et. al 2011; doi 10.1261/rna.2448911)

p13) E-site tRNA could alone be an artifact of the preparation method and hoarding of tRNA on the ribosomes for immediate usage upon restoration of active growth is very speculative since the substrate for translation is the aminoacylated tRNA bound to EF-Tu-GTP.

-In Figure 1 it would be nice to know how the agarose and SDS gels were stained.

-Are the rRNA visualized on agarose gels without prior phenol extraction of the fractions?

-For Figure 1e you state that the addition of RafH to the in vitro translation system inhibits translation. My old biochemistry teacher taught me, that if the addition of mercury, or any other ugly substance could do the same, then you need a control. Here I could suggest the His-tag antibody and hope that it can inhibit the action of added RafH and reestablish translation.

-For the sake of reproducibility, please state the number of g's at which centrifugation was done. It is missing in a few places.

-Is it really true that the RafH was purified in the presence of 50µg/ml proteinase K?

-Numbers exceeding 99,999 have a wrong format: 1,00,000 instead of 100,000 or 12,00,000 instead of 1,200,000.

Michael Askvad Sørensen, UCPH

<https://orcid.org/0000-0001-8931-2999>

Response to reviewer comments

We thank the reviewers for spending precious time reading our manuscript and providing valuable insights. We have addressed the reviewers' comments by performing additional experiments and analysis. The details are given in the response to each point. We believe that it will further increase the overall quality of the manuscript.

New information/modifications in the revised manuscript are highlighted in blue font.

Reviewer #1 (Remarks to the Author):

The article by Kumar *et al.*, reports the first cryo-electron microscopy (cryo-EM) structure of *Mycobacterium smegmatis* RafH hibernation promotion factor (HPF) bound to the ribosome at a near-atomic resolution. Overall, and despite a manuscript carelessly edited, the experiments are handled with care and the results are original. They notably explain why this specific HPF triggers ribosome hibernation of 70S monosome and not 100S disome.

Major comments:

The introduction is too long and would benefit from a shortening in order to focus more on hibernation in Mycobacteria. Covid and epidemics are certainly out of scope for this structural work.

Response: The reviewer #2 has also made similar suggestions. We appreciate these suggestions. The first two paragraphs are removed, and an introduction to ribosome hibernation is now added. Introduction to tuberculosis is shortened.

Moreover all the results are described with a lot of details at the end of the abstract and of the introduction. This weakens the following chapters of the manuscript.

Response: The abstract is now shortened to ~190 words, and only the most important findings are mentioned. Similarly, in the last paragraph of introduction, details of the findings are removed.

Results: the authors dissociate and reassociate 70S ribosomes in order to "remove the co-purified P-site bound tRNA". This step is indeed important to remove tRNAs, but also mRNA, translation factors etc. from the ribosomes. Therefore, why do the authors focus on P-site tRNAs?

Response: We agreed, it was not properly described. Definitely, the purpose was to remove all co-purified translation factors. Now the "remove the co-purified P-site bound tRNA" is replaced with "remove the co-purified translation protein factors, mRNA and tRNAs." Page no 6 Line no 132.

Indeed, they then observe (see below) tRNAs in the E-site and try to interpret this presence as a strategy to hoard tRNAs (page 13).

Response: As also commented by reviewer #2, "hoarding is very speculative." We fully agreed with the reviewers. The 'tRNA hoarding' may not be an appropriate term, as our complex is *in-vitro* reconstituted. Therefore, the term hoarding is removed from the discussion section.

The fact is that the authors use a high MgCl₂ concentration (20mM) that could trap E-site tRNAs that were also supposed to be removed. Please comment.

The high MgCl₂ concentration for *M. smegmatis* ribosome purification has been used earlier also by Hentschel *et al.*, 2017 (20 mM MgCl₂) and Mishra *et al.*, 2018 (25 mM MgCl₂). We have dissociated ribosomes in 1 mM MgCl₂ concentration. Separated subunits in sucrose density gradient centrifugation (SDGC) in 1 mM MgCl₂, and then reconstituted ribosome RafH complex in 20 mM MgCl₂. Basically, during SDGC in low MgCl₂, the E- site tRNA could have fully dissociated. However, we observed that nearly 25% ribosome population still retains the E- site tRNA. Indicating that the initial high MgCl₂ might not have influenced the E- site tRNA binding.

How do you explain the presence of E-Site tRNA after dissociation / purification and reassociation and is it compatible with RafH activity model ?

Response: The presence of E- site tRNA was a surprise for us. The ribosomes were dissociated, and subunits were purified in low (1mM) MgCl₂ concentration. The presence of E- site tRNA in hibernating ribosomes of *M. smegmatis* was earlier reported by Mishra *et al.*, 2018. The authors have purified ribosome from the stationary phase, and found that the MPY, and E- site t-RNA co-purified with the 70S ribosomes. Essentially, in a native hibernating ribosome with MPY and E- site tRNA. The presence of E- site tRNA is also reported with co-purified HPF^{long} hibernating 100S ribosome (Metzov *et al.*, 2017). A comparative analysis of our structure with the MPY and E- tRNA bound structure (Mishra *et al.*, 2018) is now added in **Supplementary Fig. S11a**. The superimposition these two structures showed that E- tRNA binds in the same conformation and its anticodon stem-loop makes interaction with the linker region of the HPF **Supplementary Fig. S11a**. The E- site tRNA interaction with the LSU is illustrated in **Supplementary Fig. S11b**. The tRNA binds to the conserved E- site binding pocket and makes similar interactions as reported in *Thermus thermophilus* by Selmer *et al.*, 2006. The presence of E- site t-RNA in ~nearly 25% of ribosomes has been explained through the structural basis for E- site tRNA's association with LSU **Supplementary Fig. S11b**, as tRNA makes extensive interaction with the LSU. A detail is provided in **the text line 286 – 308** of result section. Further, we cannot

rule out the possibility of some trace of 70S ribosome 50S ribosome pool fractions after subunit separation in SDGC (**Fig. 2b**).

Further, RafH binds in the same conformation with or without E- site tRNA, suggesting that E-site tRNA binding does not influence the RafH binding. E- site tRNA may provide additional stability to the hibernating ribosomes.

What about an mRNA into the ribosomal path?

Response: If we understood correctly, the reviewer would like to know about the fate of natively bound mRNA. RafH N- terminus domain and linker region binding overlap with the mRNA binding site (**Fig. 5a**). We showed that RafH also binds to the 30S ribosomal subunit alone (**Supplementary Fig. S2**). We believe that RafH binds to the empty ribosome.

In an earlier attempt of RafH 70S ribosome complex formation (without dissociation and re-association), we found the majority of the ribosomes with mRNA, P-, E- tRNAs in our cryo-EM map (data not shown). Suggested that RafH may not be capable of kicking out translation factors. Therefore, we changed our strategy to dissociation and re-association), of ribosomes before complex formation.

The authors use RafH protein from *M. smegmatis* in order to keep the complex with 70S homologous. However, a comparison (sequences, 3D models) with *M. tuberculosis* RafH would be useful.

Response: This is a very important suggestion. Now we have added a structure comparison of *M. smegmatis* RafH with AlphaFold predicted structure of *M. tuberculosis* RafH in **Supplementary Fig. S7** and sequence comparison in **Supplementary Fig. S8**. In **Supplementary Table 2**, which lists the *M. smegmatis* RafH interacting residues with 16S rRNA nucleotide, now the corresponding amino acid/nucleotide in *M. tuberculosis* 16S rRNA and RafH are listed in bracket. A comparison is added to the text in lines no 258 – 272 of the result section.

Minor comments

Fig 1f has no scale bar.

Response: Scale bar is now added to Fig. 1f.

Fig6 is too big for such a simple sketch.

Response: The size of Fig 6. is reduced to dimension 9 X 23 cm (WXH).

Page 7, top: « signal subtraction »: of what?

Response: it was not properly mentioned. The text is now revised. We mean; partial signal subtraction from cryo- EM electron density which corresponds to the RafH CTD and its interacting partners, bS1 r-protein and 23S rRNA H54a (**Supplementary Fig. S3b**). Now the text is revised in lines no. 165 – 168.

Page 7: "cryo- EM mass": replace by "mass" by "electron density"

Response: "mass" is replaced with "electron density" throughout the manuscript.

Page 7: "These 3 maps, of ribosome RafH complexes". Use rather "consensus map"

Response: We have changed it to "consensus map" uses, line no. line 179.

Page 9: "The signal for RafH CTD and its surrounding regions, H54a, bS1, and uS2, was subtracted". It is the contrary, isn't it?

Response: Indeed, it's contrary, and the statement is revised now. "A partial signal subtraction of cryo- EM electron density corresponding to the RafH CTD and its interacting partners bS1, H54a from polished particles were carried out". Line no. 599 -601.

Reviewer #2 (Remarks to the Author):

The manuscript "Cryo- EM structure of the mycobacterial 70S ribosome in complex with ribosome hibernation promotion factor RafH, reveals the unique mode of mycobacterial ribosome hibernation" by Niraj Kumara, Shivani Sharma & Prem S. Kaushala reports on high resolution molecular structures of in vitro formed complexes between ribosome subunits purified from *Mycobacterium smegmatis* and His-tagged RafH expressed in, and purified from, *Escherichia coli*. RafH is a hibernation promotion factor expressed during hypoxia as a member of the DosR regulon in Mycobacterium. A structure of this complex has not been reported earlier and the CTD of RafH was found to bind to a hitherto unique position (PBC) at the 30S subunit.

I have to state that I am not an expert in cryo-EM and that I am unable to evaluate the technical details of these experiments reported here. So to that part I can only say that the structure figures are clear and well explained.

I have two major concerns with this manuscript

The first two paragraphs of this manuscript contains facts about tuberculosis that is, in my view, unrelated to the scientific part of the work presented. Clearly, motivation and funding is important but I think this is overwhelming.

Response: As already mentioned in response to reviewer #1, the first two paragraphs are removed, and an introduction to ribosome hibernation is now added. Introduction to tuberculosis is shortened.

Further, along the same road, I also think the use of Mtb as an alias for *M. smegmatis* (e. g. p12) is a little dishonest, because Mtb means *M. tuberculosis* in my dictionary. In general you try to broaden your conclusions to all mycobacteria but why is *M. tuberculosis* a pathogen and *M. smegmatis* not? There must be differences and if I were mean I could suggest it is because *M. tuberculosis* is better at coping with the hypoxia in macrophages.

Response: This is important point raised and fully agreed with the reviewer. The text is now revised, and the claim is generalized by stating. Lines no 388 -390.

I would have used some of the space in the introduction to make a figure (maybe like Fig.S6 only highlighting homology) comparing known HPFs found at ribosomes of *M. smegmatis* to HPFs found in Mtb and then at various conditions. I think that would make the paper more readable for the general reader and maybe point clearer to the new findings and conclusions of this paper.

Response: We agreed, a Figure (**Supplementary Fig. S1**) is now added and referred in the introduction, which highlights the domain architecture of known HPF and its mode of hibernation, 70S or 100S. The figure is referred to in the introduction text lines no. 68 – 75.

The second major concern is: has the in vitro formed complexes used here anything to do with in vivo complexes?

Response: To address the reviewer's concern, we compared our structure with the earlier reported structures of *M. smegmatis* hibernating ribosomes, purified from Zinc ion starved condition (Li *et al.*, 2018) and stationary phase (Mishra *et al.*, 2018). Both structures have natively co-purified HPF, the MPY (**Supplementary Fig. S10a**). We found that all three structures adopt a similar conformation. The SSU is in unrotated close conformation, with an RMSD of 0.85 to 1.0 Å between phosphate backbone atoms of 16S rRNA (**Supplementary Fig. S10b**). Further, the ribosomal SSU adopts a similar conformation, unrotated close form, during initiation complex formation (**Supplementary Fig. S10b**). A similar conformation was also observed in the *M. tuberculosis* 70S ribosomal SSU (**Supplementary Fig. S10b**). It suggests that the *in-vitro* reconstitution in our structure might have negligible artefacts. A detailed description is now added to the result section lines no. 273 – 185.

I understand that the uniform formation of a high number of identical complexes promotes a high-resolution cryo- analysis. However, I begin to worry when I learn that some of the ribosomal proteins are specific for the state of growth (or stress) in *M. smegmatis* like Zn

depletion (your ref. 31; doi/10.1073/pnas.1804555115). In the present work you are purifying ribosomal subunits from a mid-log phase growing culture and combine them with a RafH protein, His-tagged and purified from *E. coli*. This is somehow fair, but you should at least discuss the possible pitfalls in this methodology when you try to broaden your conclusions as far as you do.

Response: The possible pitfalls are now discussed before the final conclusion in the second last paragraph of the discussion. However, we also did a structural comparison with the earlier reported structure *M. smegmatis* hibernating 70S ribosome, having natively co-purified HPF, MPY, by Li *et al.*, 2018 and Mishra *et al.*, 2018, 30S initiation complex by Hussain *et al.*, 2016 and *M. tuberculosis* by Yong *et al.*, 2017. In all the structures, the 30S subunit adopts a similar 'unrotated close' conformation (**Supplementary Fig S10**). The His-tag in our structure is disordered, maybe it's dynamic and not making any stable interaction with the ribosome.

In your reference 37 (DOI 10.1074/jbc.M112.364851) they purify ribosomes from hypoxia stressed *M. smegmatis* cultures and find along RafH also MSMEG_1878 (alias MPY) bound to the ribosomes, but since it is an ensemble analysis no details are available if the factors bound to the same ribosomes together or if they bound alone to different ribosomes.

Response: The NTD of RafH and MPY bind to the overlapping region on the ribosome (**Supplementary Fig. S10**) therefore, at a time, only one factor would bind to the ribosome.

Minors

p4 last paragraph) RafH is described to be overexpressed during hypoxia stress as a member of the DosR regulon. Overexpression is a strange word. I guess you mean that expression is increased?

Response: The 'overexpression' is now replaced with the 'the expression is upregulated' through.

p13) Doesn't inhibition of translation by blocking the anti-SD demand that RafH can bind 30S subunits alone? I assume that the 30S containing pre-initiation complex is expected to initiate translation on SD-bearing mRNAs by pairing between SD and anti-SD. Later the 50S subunit joins and translation begins. Is there any evidence of RafH being able to bind to 30S alone?

Response: To the best of our knowledge, there is no direct evidence. However, Trauner *et al.*, 2012 has shown that in a Δ dosR strain (RafH is DosR regulated) the degradation of 30S subunit is more compare to the wild type, and an extra copy of the *rafH* gene rescues 30S from degradation. Therefore, we performed the sucrose pelleting assay for RafH with

ribosomal 30S subunit a similar way to that of RafH with 70S ribosome. We found that the RafH is pelleting with the 30S subunit (**Supplementary Fig S2**), suggesting that the RafH interacts with the 30S subunit alone also.

Instead of this rather hypothetical intervention of translation initiation I think it could be worth considering if, and how, RafH could stabilize rRNA by its binding. It could probably be the most prominent function of RafH in vivo. In (DOI 10.1074/jbc.M112.364851) they find very unstable rRNA in a Delta(dosR) strain of which the phenotypes are reverted when ectopic rafH is expressed from a plasmid. A hint could be that in *Escherichia coli*, the degradation of rRNA begins when the subunits are free, with a RNasePH attack in 16S RNA in the anti-SD end. (Basturea et. al 2011; doi 10.1261/rna.2448911).

Response: We highly appreciate this suggestion. While reading the article suggested by the reviewer (Basturea et. al 2011), we came across some relevant references (Zundel *et al.*, 2009; Sulthana *et al.*, 2016; Prossliner *et al.*, 2021 and Lipońska & Yap 2021). We found evidence that HPF protects the ribosomes from exonuclease and endonuclease by blocking the access of its target sites on the ribosome. Further, We did a modeling study based on the literature available and added figures, **Fig. 5b** and **Supplementary Fig. S12**, and added the relevant text in the introduction, result, and discussion sections.

p13) E-site tRNA could alone be an artifact of the preparation method and hoarding of tRNA on the ribosomes for immediate usage upon restoration of active growth is very speculative since the substrate for translation is the aminoacylated tRNA bound to EF-Tu-GTP.

Response: A similar point was raised by reviewer #1. We have addressed this point while addressing reviewer #1 concern.

-In Figure 1 it would be nice to know how the agarose and SDS gels were stained.

Response: The agarose gel was stained with 0.2µg/ml Ethidium bromide, and SDS gel was stained with Coomassie blue. The same information is now added to the Methods section lines no 482 – 484 and 494 - 495 and Figure 1 legend.

-Are the rRNA visualized on agarose gels without prior phenol extraction of the fractions?

Response: Yes, fractions corresponding to respective peaks were directly run onto 2% agarose gel having 0.06% bleach. The information is now added to the Methods section 482 – 484 and 494 – 495 and Figure 1 legend. A similar procedure was earlier carried by Schubert *et. al.*, 2020. *Nat Struct Mol Biol* 27, 959–966.

-For Figure 1e you state that the addition of RafH to the in vitro translation system inhibits translation. My old biochemistry teacher taught me, that if the addition of mercury, or any other ugly substance could do the same, then you need a control. Here I could suggest the His-tag antibody and hope that it can inhibit the action of added RafH and reestablish translation.

Response: We agreed, this is a very valid point. We looked into the literature and found that point mutant and/or ribosome-targeting antibiotics were used as control (Schubert *et al.*, 2020, PMID: 32908316; Li *et al.*, 2015 PMID: 26299947;). We thought of using His-tag antibody. However, there is a chance of non-specific binding of His-tag antibody; moreover, we could not find any literature on this.

We decided to make the point mutants and created two tryptophan (W) mutants, W96A and W111A. Both, W96 and W111 make stacking base stacking interactions with the 16S rRNA nucleotides, G673 and A1518, respectively (**Fig. 2d and Fig. 3**). The *in-vitro* translation assay was repeated for RafH (Wild type), its mutants W96A and W111A and 30S targeting antibiotic, Spectinomycin (**Fig. 1e**). We found that W96A showed slightly lower inhibition whereas, W111A showed similar inhibition compared to the wild type. Spectinomycin showed a similar inhibition at 5X higher molar concentration to that of the ribosome. The difference in the inhibition between the two mutants may be because of their strategic location of interaction with 16S rRNA. As W111A is immediately surrounded by other interacting residues, R109, E110, and R112 (**Supplementary Table S2**). Maybe these interactions compensate for the loss of base stacking interaction in the W111A mutant. Text is added to result and method section.

-For the sake of reproducibility, please state the number of g's at which centrifugation was done. It is missing in a few places.

Response: All the centrifugation earlier reported in RPM is now replaced with corresponding x g values in the Method section.

-Is it really true that the RafH was purified in the presence of 50µg/ml proteinase K?

Response: Sorry, it was a typo. We have used Protease Inhibitor Cocktail (Roche). The same is updated in the method section. Line 511

-Numbers exceeding 99,999 have a wrong format: 1,00,000 instead of 100,000 or 12,00,000 instead of 1,200,000.

Response: The format is now corrected, throughout.

REVIEWERS' COMMENTS

Reviewer #1 (Remarks to the Author):

The authors have carefully responded to each of my concerns, which has significantly improved the overall reading of the manuscript. I particularly appreciate that they did explain the presence of E-site tRNAs in their structures and performed structural comparisons of *M. smegmatis* RafH with predicted structure of *M. tuberculosis* RafH.

Reviewer #2 (Remarks to the Author):

The revision of the manuscript "Cry-EM structure....." by N. Kumara, S. Sharma and PS. Kaushala has improved the clarity and readability of the work.

Minor things:

Line 117: are some words missing?

Line 146: Here you introduce experiments with two mutant forms of RafH without any explanation about why - and without an explanation about why exactly these two mutant forms. I think your arguments from the rebuttal letter would make it clearer. Unfortunately, both mutant forms inhibit nearly as well as the wt.

Figure 1e: Something is wrong with the scale on the Y-axis. 1×10^6 occurs twice and there seems no reason to make the scale discontinuous. Also, the unit "RLU" is never explained but after reading "Methods" one could guess. May I suggest something like "Translational activity (RLU)" and a further explanation in the caption that production rate of nLuc activity is measured?

Best regards,

Michael A. Sørensen

Response to reviewer comments

The reviewer comments are addressed in the revised manuscript, and highlighted in red font. We hope that the comments were addressed to the reviewer's satisfaction.

Reviewer #1 (Remarks to the Author):

The authors have carefully responded to each of my concerns, which has significantly improved the overall reading of the manuscript. I particularly appreciate that they did explain the presence of E-site tRNAs in their structures and performed structural comparisons of *M. smegmatis* RafH with predicted structure of *M. tuberculosis* RafH.

Reviewer #2 (Remarks to the Author):

The revision of the manuscript "Cry-EM structure....." by N. Kumara, S. Sharmaa and PS. Kaushala has improved the clarity and readability of the work.

Minor things:

Line 117: are some words missing?

Some of the words were missing. Now the sentence is complete.

Line 146: Here you introduce experiments with two mutant forms of RafH without any explanation about why - and without an explanation about why exactly these two mutant forms. I think your arguments from the rebuttal letter would make it clearer. Unfortunately, both mutant forms inhibit nearly as well as the wt.

Now the text is added in the result section line no 149-150 and method section line no 527-528. The W96A shows slightly lower inhibition, which may be because of its strategic binding site at the decoding center of the ribosomal small subunit.

Figure 1e: Something is wrong with the scale on the Y-axis. 1×10^6 occurs twice. The scale was shown without after decimal points, and the values were 1.3×10^6 and 1.6×10^6 .

and there seems no reason to make the scale discontinuous.

Figure 1e is revised with a continuous scale.

Also, the unit "RLU" is never explained but after reading "Methods" one could guess. May I suggest something like "Translational activity (RLU)"

RLU, Relative Luminescence Unit, is now explained in the method and also added to Figure Legend 1. To keep the consistency with the literature, we would prefer to keep, RLU (Relative Luminescence Unit).

and a further explanation in the caption that production rate of nLuc activity is measured?

Explanation is now added in the caption.